

# Interactions between chaperone and energy storage networks during the evolution of *Legionella pneumophila* under heat shock

Jeffrey Liang and Sebastien P. Faucher

Department of Natural Resource Sciences, McGill University, Sainte-Anne-de-Bellevue, Quebec, Canada

## ABSTRACT

Waterborne transmission of the bacterium *Legionella pneumophila* has emerged as a major cause of severe nosocomial infections of major public health impact. The major route of transmission involves the uptake of aerosolized bacteria, often from the contaminated hot water systems of large buildings. Public health regulations aimed at controlling the mesophilic pathogen are generally concerned with acute pasteurization and maintaining high temperatures at the heating systems and throughout the plumbing of hot water systems, but *L. pneumophila* is often able to survive these treatments due to both bacterium-intrinsic and environmental factors. Previous work has established an experimental evolution system to model the observations of increased heat resistance in repeatedly but unsuccessfully pasteurized *L. pneumophila* populations. Here, we show rapid fixation of novel alleles in lineages selected for resistance to heat shock and shifts in mutational profile related to increases in the temperature of selection. Gene-level and nucleotide-level parallelisms between independently-evolving lineages show the centrality of the DnaJ/DnaK chaperone system in the heat resistance of *L. pneumophila*. Inference of epistatic interactions through reverse genetics shows an unexpected interaction between DnaJ/DnaK and the polyhydroxybutyrate-accumulation energy storage mechanism used by the species to survive long-term starvation in low-nutrient environments.

# INTRODUCTION

Portions of this text were previously published as part of a preprint (*Liang & Faucher, 2023*).

One of the major challenges facing the managers of large buildings is the problem of pathogen growth in hot water distribution systems (HWDS) (*USEPA, 2016*). Waterborne transmission is a vector for many worrying pathogens; *Pseudomonas* spp., non-tuberculous mycobacteria, and *Legionella pneumophila* tally amongst the most epidemiologically concerning (*Greco et al., 2020*). *Legionella* infection alone was estimated to cause over 1,000 deaths in 2014, imposing direct healthcare costs of USD $400 million

Corresponding author
Sebastien P. Faucher,
sebastien.faucher2@mcgill.ca

(*Collier et al., 2021*; *DeFlorio-Barker, Shrestha & Dorevitch, 2021*). Consequently, public health authorities impose regulations on the management of these systems aimed at lowering the risks of disease transmission, especially in hospitals with vulnerable patients at risk of nosocomial infection (*Almeida et al., 2016*; *USEPA, 2016*). While certain interventions, such as UV exposure or point-of-use filtration systems, are directly meant to interfere with the transmission of the pathogen to the patient, the greater part of these interventions are directed at clearing the contaminating populations from within these HWDS (*Lin, Stout & Yu, 2011*; *Almeida et al., 2016*). Certain protocols disinfect the water supply through chemical means, but thermal control regimes are widely used for both steady state suppression of bacterial populations and for intensive remedial treatment when outbreaks occur (*Rhoads et al., 2015*; *USEPA, 2016*).

Interactions between physical and biological factors in HWDS provide *L. pneumophila* a hospitable niche for replication. These plumbing systems can sustain biofilm accumulations and populations of susceptible protozoa that support the growth of infectious bacteria (*Rowbotham, 1980*; *Gomez-Valero, Rusniok & Buchrieser, 2009*). The adaptations that allow *L. pneumophila* to infect permissive environmental eukaryotes have also potentiated its virulence in human hosts, allowing aerosolized bacteria to survive and replicate inside alveolar macrophages when inhaled (*Sadosky, Wiater & Shuman, 1993*; *Park, Ghosh & O'Connor, 2020*). During its life cycle as an environmental bacterium, *L. pneumophila* alternates broadly between a replicative phase as an intracellular parasite of free-living phagocytic eukaryotes and a transmissive phase as a planktonic cell in search of new hosts to infect (*Molofsky & Swanson, 2004*; *Schell et al., 2016*). Adaptations to freshwater environments have left the species well-adapted to the development of new niches within water infrastructure, including cooling towers and plumbing systems. Bacterial proliferation in these contexts provides *L. pneumophila* easy dispersion pathways which intersect dangerously with the humans living adjacent to these systems (*USEPA, 2016*; *Bédard et al., 2019*). The most clinically severe manifestation of *Legionella* infection is known as Legionnaires' disease and presents as a severe pneumonia, often in tandem with gastrointestinal and neurological symptoms (*Cunha, Burillo & Bouza, 2016*). The mortality rate of Legionnaires' disease is estimated at around 9%, with incidence rates rising in recent years (*Centers for Disease Control & Prevention, 2017*; *Collier et al., 2021*). Immunosuppressed individuals are at higher risk of infection dissemination or relapse, and also suffer higher mortality rates (*Cunha, Burillo & Bouza, 2016*).

As the consequences of uncontrolled *Legionella* growth in hot water systems are severe, a body of research has accumulated concerning the short-term and long-term efficacies of anti-*Legionella* control measures (*Allegra et al., 2011*; *Epalle et al., 2015*; *Bédard et al., 2016a*, *2021*). With the necessary infrastructure often already in place, thermal disinfection (often termed superheat-and-flush) treatments are a common first-line response to outbreaks or when *Legionella* population levels reach regulatory thresholds (*Lin et al., 1998*; *Chen et al., 2005*). Implementation can vary depending on the topology of the plumbing network and situation of hot water heaters, but the common intention is to guarantee that the temperature of the entire hot water distribution network from water heater through to distal taps is lethal to *L. pneumophila* (*Lin et al., 1998*). Unfortunately, a

number of healthcare facilities have reported that the remedial effects of hot water treatment are only temporary, caused by bacterial regrowth after the acute heat shock subsides (*Allegra et al., 2011*; *Bédard et al., 2016b*).

Intricate interdependencies of structural systems and regulatory networks in the bacterial cell form a vast attack surface for heat shock's non-specific physicochemical trauma. The lethal effects of heat shock are thought to involve membrane destabilization, nucleoid disruption, and, most prominently, toxic protein unfolding and aggregation (*Mitsuzawa, Deguchi & Horikoshi, 2006*; *Murata et al., 2011*; *Li & Gänzle, 2016*; *Lv et al., 2023*). Accordingly, the primary bacterial heat shock response co-ordinates the action of an arsenal of chaperones, refoldases, and disaggregases to repair protein damage, as well as a collection of proteases to degrade irreversibly destabilized polypeptides (*Gamer et al., 1996*; *Genest et al., 2011*). A number of cross-interactions have been demonstrated between the heat shock response and regulatory pathways involved in virulence development and metabolic switching. In *L. pneumophila*, chaperone protein HtpB is a major surface-exposed antigen with host-directed trafficking manipulation activity that helps the bacteria survive early infection (*Hoffman, Butler & Quinn, 1989*; *Hoffman et al., 1990*; *Chong et al., 2009*). Similarly, the AAA+ protease ClpP is involved in balancing cellular concentrations of the CsrR regulatory protein across the replicative and transmissive phases (*Ge et al., 2019*) while reciprocally, genes involved in the stringent response and with two-component signaling are pleiotropically involved in heat shock resistance (*Mendis et al., 2018*; *Saoud, Mani & Faucher, 2021*; *Liang & Faucher, 2022*).

Complex phenotypes like heat shock resistance are well-suited for study through adaptive laboratory evolution, a process which allows the model organism to explore a multiplicity of adaptive paths towards locally optimal genotypes (*Blaby et al., 2012*; *Dykhuizen, 2016*). Adaptive laboratory evolution has been adopted as a powerful research tool to examine the evolved responses to a barrage of experimental conditions including ionizing radiation, oxidative stress, and non-standard growth temperatures (*Tenaillon et al., 2012*; *Bruckbauer et al., 2020*; *Rodríguez-Rojas et al., 2020*). Developments in sequencing technology, in particular, now allow researchers to identify the genotypes of sequenced isolates and allele frequencies of sequenced populations and deduce the molecular bases behind expressed phenotypes (*Bruger & Marx, 2018*).

The foundation of this study is an adaptive laboratory evolution model designed to replicate the effects of thermal disinfection in a *bona fide* building water system. We simulated the pressures of transient hot water demand and incomplete pasteurization cycles by subjecting our laboratory lineages to short bursts of high temperature. This process was successful in generating heat-adapted lineages which could tolerate exposure to higher temperatures without trade-offs in axenic or intracellular growth potential (*Liang, Cameron & Faucher, 2023*). This present work focuses on charting the evolutionary trajectories of three heat-adapted lineages through a time-transect of their adaptation to heat shock and presenting the alleles which promote their heat resistance. Notably, while many acquired mutations affected canonical heat shock proteins, one of the mutations of greatest effect, which arose early on in two heat-adapted lineages but none of the control lineages (*Liang, Cameron & Faucher, 2023*), targeted a phasin protein interacting with

carbon and energy storage mechanisms in the bacterial cell. We show unexpected epistatic interactions between this mutation and the *dnaJ/dnaK* chaperone system, highlighting connections between the life phase switching of *L. pneumophila* and its capability to tolerate heat shock.

## MATERIALS AND METHODS

### Strains and culture conditions

CYE (ACES-buffered charcoal yeast extract) agar plates adjusted to pH 6.9 and supplemented with 0.25 g/L L-cysteine and 0.4 g/L ferric pyrophosphate, as standard, were used for the routine culture of *L. pneumophila* throughout this study. Strains were streaked from −80 °C storage stocks to isolation and grown for 72 h at 37 °C. Broth cultures were grown in AYE, a liquid medium identical in composition to CYE but lacking charcoal and agar. Unless otherwise indicated, *L. pneumophila* cells were harvested from post-log phase broth culture, rinsed three times in Fraquil, a simulated North American freshwater medium of defined composition (*Mendis, McBride & Faucher, 2015*), and suspended to a final $OD_{600}$ of 0.1 in Fraquil for 24 h to stimulate a shift to a nutrient-deprived transmissive phase. Where necessary, chloramphenicol was used at a concentration of 5 µg/mL, gentamycin was used at a concentration of 10 µg/mL and kanamycin was used at a concentration of 25 µg/mL.

### Adaptive laboratory evolution and population selection

In a previously-published adaptive laboratory evolution, replicate lineages of *L. pneumophila* were adapted to heat shock (described in detail in *Liang, Cameron & Faucher, 2023*). In brief, a common ancestor strain–clinical isolate Philadelphia-1 ATCC 33152–was split into six replicate HA-lineages to experimentally promote adaptation to increased temperatures and six replicate C-lineages as controls without selection for survival under heat. Experimental passages were conducted 70 times, during which the temperatures of exposure were gradually raised in tandem with the HA-lineages' increase in heat tolerance.

Frozen samples were collected after every passage during the experimental evolution procedure, which allowed us to query the tempo and dynamics of mutation acquisition in this experiment. We sequenced one population sample of each of lineages HA-1, HA-2, and HA-3 after every five passages–a total of 42 samples–based on our estimates of acceptable sequencing depth. Frozen stocks were grown directly in 1 mL AYE overnight (with freedom from contamination ascertained by streaking the dense culture on CYE plates) to capture the population diversity of each sample. Genomic DNA was collected using the Wizard Genomic DNA Collection Kit (Promega, Madison, WI, USA) and stored at −20 °C before submission to the McGill Genome Center for library preparation and sequencing on an Illumina MiSeq platform using a 600-cycle v3 MiSeq Reagent Kit.

Reads were trimmed and filtered for quality using Fastp v0.20.1 (*Chen et al., 2018*) and aligned to the genome of the ancestor strain (CP015927.1) with Breseq 0.36.1

(*Deatherage & Barrick, 2014*) using default polymorphism settings and subprograms R 4.1.2 and bowtie2 2.4.1 (*Langmead & Salzberg, 2012*; *R Core Team, 2021*). Overall, reads aligned at a 43.5× coverage across the *L. pneumophila* genome (max = 57.78, min = 27.81). False positive polymorphisms were observed around homopolymeric runs, with mutations showing elevated strand bias by Fisher's Exact Test being removed from further analysis. To estimate the speed at which mutations accumulated over the time course of the adaptive laboratory evolution, mutation load was calculated as a sum of the frequencies of polymorphic and fixed derived alleles in the sequenced populations at each five-passage interval, excluding the two mutations known to be fixed in our laboratory's strain of Philadelphia-1. These data were used to plot a linear regression with a constraint of zero mutations at the start of the experiment.

## Mutant selection and construction

Despite repeated attempts to construct scar-free deletions in Philadelphia-1 using a *pGEM-kan-mazF* resistance/suicide cassette (*Bailo et al., 2019*), we were unable to successfully transfer the derived alleles in isolation or combination into the ancestral background. Accordingly, we adapted our standard allelic exchange protocol for the construction of knock-out mutations to shuttle the derived polymorphisms into KS79, a genetically tractable descendant of Philadelphia-1, as our wild-type background. Constructing fragments by overlap extension PCR, we synthesized DNA products that contained two directly-adjoining 1–2 kbp chromosomal fragments, the upstream and downstream fragments, separated variously by a chloramphenicol, gentamycin, or kanamycin resistance cassette sourced from plasmids pMMB207c, pBBR1-MCS2, and pSF6 respectively (*Kovach et al., 1995*; *Chen et al., 2004*; *Faucher, Mueller & Shuman, 2011*). The fragments were individually amplified using the primers listed in Table S1: upstream fragments with primer finishing in -UF/-UR, downstream fragments with primers finishing in DF/DR and resistance cassette with primers finishing in KnF/KnR, CmF/CmR or GnF-GnR. The mutant allele was carried in either the upstream or downstream fragment of this construct, depending on the genomic context and which orientation we judged would have a milder polar effect on the operon and other surrounding genes. These fragments were transformed into KS79 by natural competence and the resulting mutants were isolated on antibiotic-selective AYE plates and confirmed by Sanger sequencing.

## Heat shock challenge

To minimize inter-sample variability resulting from artifacts of handling, we conducted heat shock experiments in 60 μL volumes in triplicate in 96-well non-skirted PCR plates (VWR) (*Liang, Cameron & Faucher, 2023*). Overnight cultures grown in AYE were rinsed three times and resuspended in Fraquil for 24 h at an $OD_{600}$ of 0.1. Samples were placed into a Veriti 96-well thermocycler pre-heated to 55 °C and left for 10, 20, or 30 min before they were actively cooled in the thermocycler to 20 °C. Samples were immediately diluted and plated on CYE plates to determine CFU/mL. Heat tolerance was quantified by calculating the thermal kill rate ($TKR_{55\ °C}$), an inversion of the D-value of pasteurization

(*Mazzola, Penna & da S Martins, 2003*), as a ordinary least squares (OLS) log-linear regression of CFU/mL and time of exposure:

$$TKR_{55\,°C} = OLS\big(\log_{10}(CFU/mL), t\big)$$

Statistical analysis was conducted in Prism 9.4.1 using an ANCOVA-equivalent two-step calculation by firstly computing TKS as above and secondly comparing regression parameters by one-way ANOVA.

## qPCR and RNA extraction

To compare transcriptomic responses between *L. pneumophila* strains expressing different alleles of *phaP*, strains grown overnight in AYE were suspended at an $OD_{600}$ of 1.0 in Fraquil for one hour at room temperature in biological triplicate. Trizol, by the manufacturer's protocol, was used to collect RNA from 1 mL samples before and after a 5-min immersion at 55 °C in a circulating water bath. We have previously published assessments of the transcriptomic response of *L. pneumophila* to heat shock under this treatment to balance the strength of the induced heat shock response against the noise of transcriptomic dysregulation in lethally stressed cells (*Liang & Faucher, 2022*; *Liang, Cameron & Faucher, 2023*). The collected RNA was treated with Dnase I and Dnase Inactivation Reagent (Thermo Fisher Scientific, Waltham, MA, USA), quantified by NanoDrop, and stored at −20 °C in nuclease-free water (Thermo Fisher Scientific, Waltham, MA, USA). Reverse transcription was performed in 20 uL at 42 °C for 1 h on 1 ug RNA samples using 10 U/uL Protoscript II (NEB) and 3 uM random hexamer primers (Invitrogen, Waltham, MA, USA) following the manufacturers' recommendations. We used previously validated primers (Table S2) for *rpoH*, *dnaJ*, *dnaK*, and *16S* (*Liang, Cameron & Faucher, 2023*) in 20 uL RT-qPCR reactions in technical triplicates in a 7500 Fast Real-Time PCR (Applied Biosystems, Waltham, MA, USA) in MicroAmp Fast Optical 96-Well Plates (Thermo Fisher Scientific, Waltham, MA, USA) using 10 uL iTaq Universal SYBR-green supermix (BioRad, Hercules, CA, USA), 500 nM primers, and 50 ng cDNA for fluorescence readout. Reactions were programmed for 30 s denaturation at 95 °C and forty cycles of 5 s denaturation at 95 °C and 30 s extension at 60 °C, and $C_T$ values were recorded from the 7500 Fast Software (Applied Biosystems, Waltham, MA, USA). Normalization between samples was applied using 16S rRNA as a common housekeeping gene and efficiency-adjusted ddCT (*Yuan, Wang & Stewart, 2008*) was used to compare expression between heat-exposed and unexposed bacteria.

## Spectrophotometric evaluation of PHB content

We applied nile red staining to post-exponential phase *L. pneumophila* cells to quantify intracellular PHB levels. Although gas chromatography methods are often used to measure total PHB content in a biomass sample, studies in *L. pneumophila* and other relevant bacteria have found good concordance between this standard and with fluorescent protocols (*James et al., 1999*; *Alves et al., 2017*). *L. pneumophila* strains were grown in AYE shaking at 37 °C to mid-log phase in preparation and used to inoculate cultures to an $OD_{600}$ of 0.1 for synchronized growth under incubation in the same conditions. A 100 μL

sample was pelleted at 5K ×*g* for 5 min and rinsed twice in 1 mL Milli-Q water. The final suspension was pelleted as before and resuspended in 1 mL ice-cold 30% ethanol in Milli-Q water and incubated at 4 °C for 30 min to permeabilize the cells and improve Nile red staining. A Nile red stock solution was prepared at 10 mg/mL in DMSO and 1 μL was added to each sample and incubated at 4 °C for 30 min. Absorbance at 600 nm and fluorescence with 535 nm excitation and 590 nm emission frequencies were measured in 1 cm cuvettes in a SpectraMax m2e spectrophotometer, and relative staining was calculated as the ratio of Nile red fluorescence to $OD_{600}$.

### dN/dS analysis

The measure of the ratio between synonymous and non-synonymous mutations (dN/dS) in protein-coding regions between multiple genomes is a common quantity used to detect adaptation or functional constraint. GenomegaMap (*Wilson & The CRyPTIC Consortium, 2020*) applies this calculation as a phylogeny-free model to generate a Bayesian estimate of within-species dN/dS. Homologous sequences to the analyzed genes were identified across a panel of 120 fully assembled *L. pneumophila* genomes downloaded from GenBank (Table S3), pruned of invariant sequence duplicates, and processed using the ALFIX-MACSE v2 (*Ranwez et al., 2018*) pipeline to produce a HMMCleaner-filtered codon-aware gene alignment. The sliding window model in genomegaMap v1.0.1 was used to estimate per-codon distribution of ω with default priors and MCMC chains of 100,000 iterations.

### Data availability

Illumina short-read data were deposited with the National Centre for Biotechnology Information's Sequence Read Archive and is available with accession PRJNA994942.

## RESULTS

### Mutation acquisition during adaptation to heat shock

Extending our research into the experimental evolution of *L. pneumophila* to model pasteurization resistance (*Liang, Cameron & Faucher, 2023*), we conducted a time-transect to observe the ebb and flow of novel alleles as they competed in our experimental lineages. Population-level sequencing of three of the lineages at five-passage intervals from the fifth passage up until the seventieth and final passage was done to detect the flow of novel alleles as their frequencies varied along the time course (Tables S4–S6, Fig. 1A). These three lineages acquired novel alleles at an overall linear rate of 0.181 fixed mutations per passage, or equivalently 0.908 per five-passage interval (Fig. 1B). We observed no marked discrepancies from this rate of mutation accumulation, implying that they evolved under equivalent evolutionary forces (*i.e.*, selection through heat shock without development of a hypermutator phenotype (*Tenaillon et al., 2016*)). Temperatures were raised at three time points over the course of the adaptive laboratory evolution experiment, resulting in four discrete challenge environments for the heat-adapted strains: 55 °C between passages 1 and 10, 57 °C between passages 11 and 30, 58 °C between passages 31 and 50, and 59 °C between passages 51 and 70. These thresholds were associated with large shifts in allele

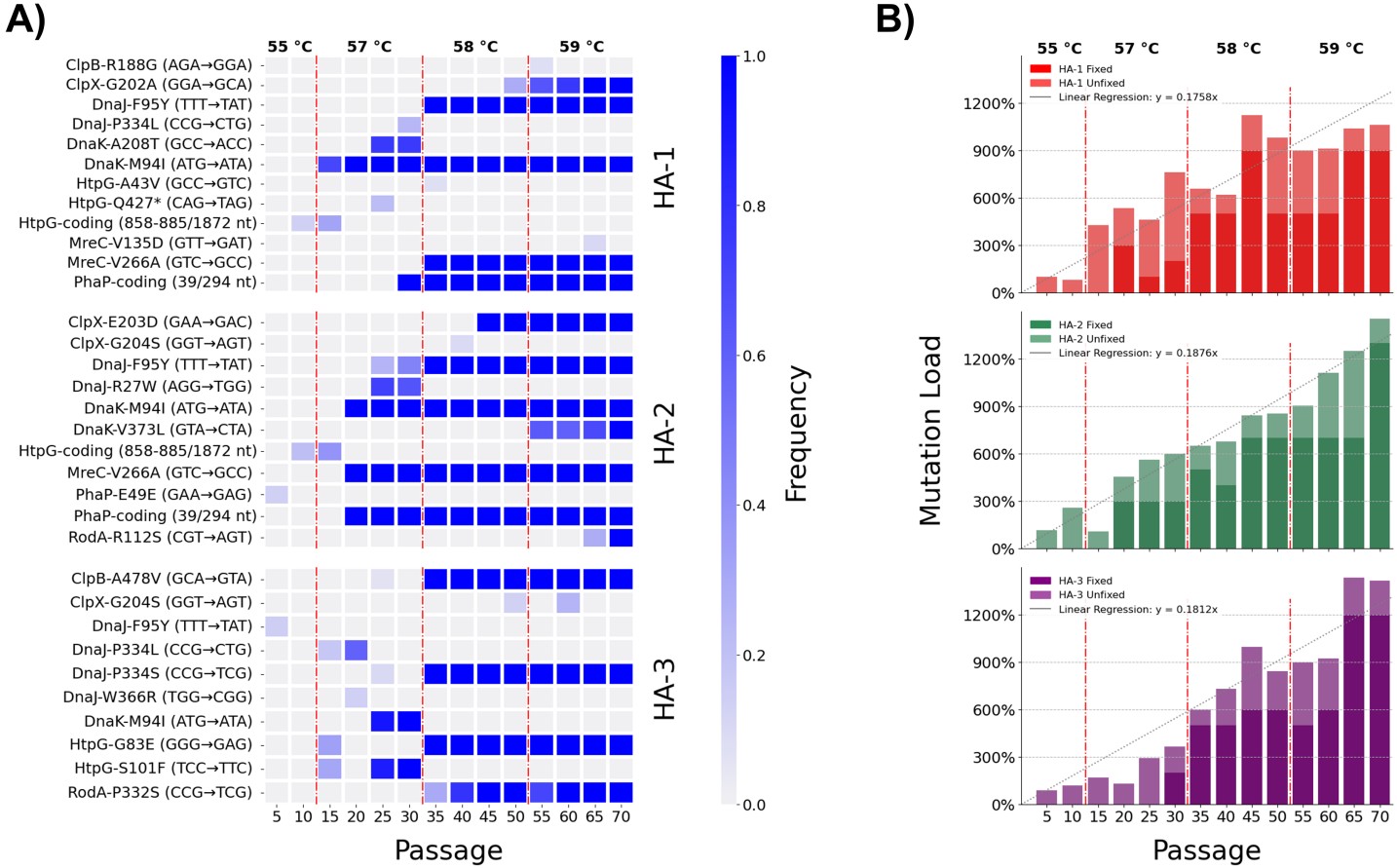

**Figure 1 Dynamics of mutation innovation and fixation in *L. pneumophila* populations undergoing experimental selection for heat shock tolerance.** (A) Allele frequencies of mutations accumulated in eight selected genes (*clpB*, *clpX*, *dnaJ*, *dnaK*, *htpG*, *mreC*, *phaP*, and *rodA*) across the three sequenced heat-adapted populations. (B) Linear accumulation of mutations in three sequenced heat-adapted (HA) lineages through 70 passages of selection. Mutation load totals fixed mutations and allele frequencies of polymorphisms annotated by Breseq 0.36.1 per-lineage. Linear fit depicts results of a linear regression of all per-lineage mutation load points with respect to passage number constrained to intercept (x, y) = (0, 0). Vertical dashed lines represent times of temperature increase.

frequency–both gains and losses of mutations–across the sequenced lineages (Fig. 1A). The 35-passage threshold, for instance, reflecting an increase in challenge temperature from 57 °C to 58 °C caused mutations DnaJ-P334L and DnaK-A208T to revert to the wild-type in the HA-1 population while mutations DnaJ-F95Y and MreC-V226A rapidly fixed.

As the diverse coterie of acquired mutations suggests, the stress of heat shock causes numerous insults to the integrity of a bacterial cell, hence the complex interactions in the bacterial heat shock response. Eight genes which were mutated in the three heat-adapted lineages under investigation were chosen for direct genetic follow-up. Five were strongly expected to be involved in the heat shock response of *L. pneumophila*, and comprise the chaperones *htpG*, *dnaJ*, *dnaK*, *clpB*, and *clpX* (UniProt accession Q5ZVS1, Q5ZTY4, Q5ZTY3, Q5ZUP3, and Q5ZUE0). Two further genes *mreC* and *rodA* (Q5ZXC0 and Q5ZVR6) were expected to be involved in cell wall synthesis and maintenance. The eighth gene was annotated as a hypothetical protein in GenBank and as a phasin_2

domain-containing protein on UniProt (Q5ZY12) which we henceforth refer to as *phaP*, in accordance with its homologues in *Cupriavidus necator* (*Pötter et al., 2004*). These genes were identified in our earlier publication as being of interest since non-synonymous mutations were fixed independently in two or more of the five heat-adapted lineages after 70 passages, including in at least two of the three lineages under consideration in this study (Fig. 1A) (*Liang, Cameron & Faucher, 2023*).

## Estimation of selection parameters

A computational inquiry was conducted into the evolutionary plasticity of these genes by querying the genomic evidence of publicly available *L. pneumophila* sequences. For each of these eight genes, the unique sequences seen across the species were aligned codon-wise allowing for the measure of $d_N/d_S$ (or $\omega$), an estimation of the balance between positive and negative selection based on the observed codon changes in protein-coding regions. Bayesian estimation of $\omega$ was conducted with genomegaMap (*Wilson & The CRyPTIC Consortium, 2020*) for each of these eight genes, as well as for three other genes of interest (*rpoB, mavF, nuoG*) observed in heat-adapted *L. pneumophila* (*Liang, Cameron & Faucher, 2023*) (Fig. S1). Average per-gene $\omega$–as well as typical per-codon $\omega$–was below 1, evidence against strong positive selection driving evolution between these sequences. In particular, the five chaperone genes had significantly lower evolutionary flexibility, despite their repeated acquisition of novel mutations in the adaptive laboratory evolution model. The chaperones (*htpG, dnaJ, dnaK, clpB, clpX*) had a significantly lower proportion of codons whose 95% equal-tailed credibility intervals encompassed $\omega = 1$ – ones whose encoded residues had relaxed mutational constraints–as compared to the six non-chaperones (*rpoB, phaP, mreC, rodA, mavF, nuoG*) (two-tailed Mann-Whitney test, $p = 0.0498$). Interestingly, the *mavF* gene had a distinctively higher estimate of $\omega$ than the other tested genes, most likely a reflection of the relatively relaxed bounds on its form and function as a member of *L. pneumophila*'s varied and redundant secreted T4SS effector corps, consistent with other effector genes (*Huang et al., 2011*; *Park, Ghosh & O'Connor, 2020*; *Zhan et al., 2021*) (Fig. S1J).

## Influence of individual alleles on heat shock tolerance

To study the effects of these mutations in isolation without possible interactions with the other acquired mutations in the heat-adapted lineages, these alleles were transferred into a wild-type genetic background. Due to the diverse cell processes impacted by heat shock, we expected resistance to arise through small stepwise gains in fitness from many contributory mutations, as opposed to a few large effect size mutations. Although we initially attempted to construct scar-free mutations in Philadelphia-1—which would have allowed for greater combinatorial flexibility when isolating and assembling these mutations—we were unable to consistently produce these mutants (*Bailo et al., 2019*). Instead, we constructed and mobilized high-fidelity PCR products amplified from heat-adapted isolate gDNA containing the derived mutations into KS79, a genetically tractable direct descendant strain of Philadelphia-1. Using selection for three antibiotics, we simulated the acquisition of one, two, or three mutations onto the wild-type background in the order that they were seen in

the evolution experiment (Tables S4–S7). Mutations were confirmed by Sanger sequencing. With this strategy, we organized the mutants into four trajectories (TRA): TRA-1 with three mutations from evolutionary lineage HA-1, TRA-2 and TRA-3 each with three distinct mutations from lineage HA-2, and TRA-4 with seven mutations from lineage HA-3. TRA-1 recapitulates the stepwise fixation in lineage HA-1 of chaperone mutations $dnaK^{M94I}$, $dnaJ^{F95Y}$, and $clpX^{G202A}$ after passages 20, 35, and 65 respectively. The mutations $phaP^{A8\to A7}$ and $mreC^{V266A}$ that were observed in lineage HA-1 were also seen in lineage HA-2; therefore, they were grouped into TRA-2 along with a $rodA^{R112S}$ seen in HA-2 on the basis of their not being canonical heat shock proteins. TRA-3 involves mutations in the same genes as in TRA-1, but with distinct alleles: $dnaK^{V373L, M94I}$, $dnaJ^{F95Y}$, and $clpX^{E203D}$. Finally, TRA-4 represents mutations observed in lineage HA-3 and, because mutations $clpB^{A478V}$, $dnaJ^{P334S}$, and $htpG^{G83E}$ were all fixed in passage 35, represents all unique single, double, and triple mutants at these three loci.

Comparisons of survivability under heat shock were drawn by evaluating the rates of population decline from a 24-h starved water-suspension of *L. pneumophila* cells during a 30-min exposure at 55 °C, the challenge temperature from the initial ten passages of the evolution procedure. This temperature was chosen as an intermediate stress which could be used to consistently compare between wild-type and partially heat resistant strains. The magnitude of the heat resistance of the evolved lineages is seen in the positive control HA-1, HA-2, and HA-3 lineages in Figs. 2A–2D, which have near-zero thermal kill rates at this temperature. The data from this system of multiple mobilization was more consistent with a multi-step model of the evolution of heat shock resistance with small benefits accruing from each of a large number of mutations. Single mutations of *phaP*, *dnaJ*, *dnaK*, and *htpG* each individually contributed significant increases in heat tolerance (Figs. 2B–2D). Two distinct mutations of *dnaK* developed in two independent lineages: $dnaK^{M94I}$ in HA-1 and $dnaK^{M94I, V373L}$ in HA-2. Single mutant TRA-1.1 ($dnaK^{M94I}$) was not significantly more heat tolerant than KS79, but the aggregate phenotype of TRA-1.3 ($dnaK^{M94I}$, $dnaJ^{F95Y}$, $clpX^{G202A}$) was a significant increase in heat tolerance. The signature of additive effects was not apparent in the other trajectories (Figs. 2B–2D), suggesting epistatic interactions between the evolved alleles. No permutation of mutations was able to capture the full magnitude of heat resistance seen in the fully evolved lineages, showing that heat resistance largely evolves as the emergent phenotype of polygenic adaptation. Previous work found, surprisingly, that *L. pneumophila* lacking *htpG*, the bacterial 90-kilodalton heat shock protein, were more able to tolerate heat shock than wild-type strains (*Liang & Faucher, 2022*). This was corroborated in our adaptive laboratory evolution model, where frameshift and nonsense mutations in *htpG* were observed in lineages not explored in this manuscript. HA-3, however, expresses non-synonymous mutation G83E improving resistance to heat shock (Fig. 2D) through an unknown but presumably different mechanism.

## Frameshifting in a phasin-like protein affects heat shock survival

The single mutation of greatest effect size did not involve a characterized heat shock protein but was a frameshift mutation, $A_8 \to A_7$, in a hypothesized phasin protein PhaP

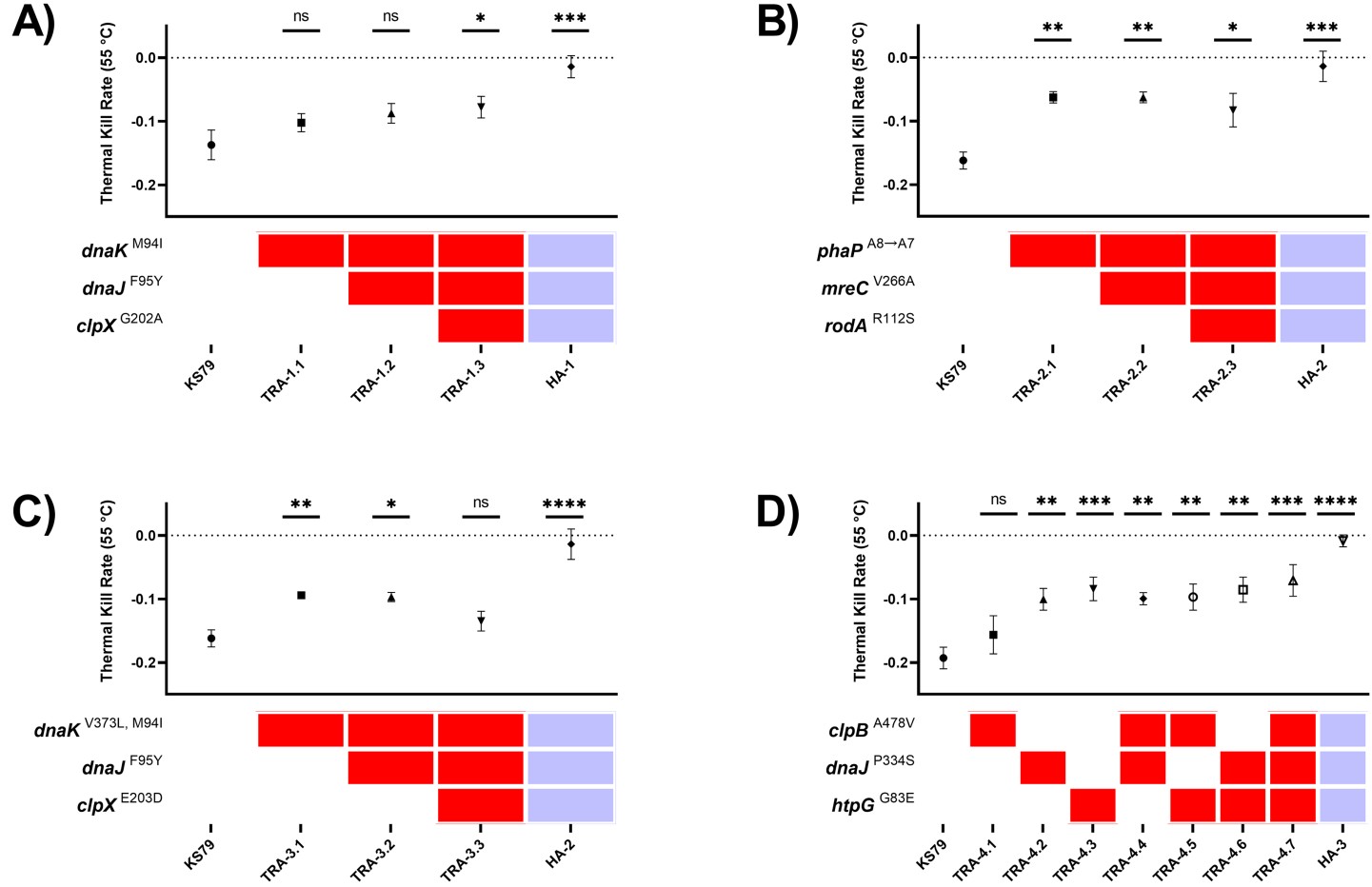

**Figure 2 Heat tolerance comparisons between mutant *L. pneumophila* strains constructed with evolutionarily-fixed mutations in isolation or combination.** (A–D) Comparisons of thermal kill rate at 55 °C of wild-type KS79 and constructed single, double, and triple mutants (identified by key below each graph), as well as the heat-adapted (HA) lineage produced by 70 rounds of selection (which contains all three mutated alleles, as indicated by key). Mutants are grouped by trajectories (TRA). See Results section for details. Data shown represent mean +/− SEM. Significance measurements depict two-tailed one-way ANCOVA relative to KS79, $n = 3$, ($^*p < 0.05$; $^{**}p < 0.01$; $^{***}p < 0.001$; $^{****}p < 0.0001$). Dotted horizontal line at y = 0 indicates full thermal resistance to 55 °C exposure.

(Fig. 2B). This deletion of an adenine base in a homopolymeric tract of eight identical nucleotides beginning at position 39 was seen in both HA-1 and HA-2 (Fig. 3A). Conversely, this poly(A) region was also the target of an $A_8 \rightarrow A_9$ frameshift mutation in HA-6 (*Liang, Cameron & Faucher, 2023*), a replicate lineage not in the focus of the present study. This frameshift occurs near the 5′ terminus of the gene which is consistent with the hypothesis that the higher rates of mutation at homopolymeric tracts is used by bacteria as a regulatory mechanism to reversibly inactivate genes or control phase switching (*Orsi, Bowen & Wiedmann, 2010*). AlphaFold (*Jumper et al., 2021*; *Varadi et al., 2022*) predicts this polypeptide to fold into two amphipathic α-helices connected by a flexible linker (Fig. 3B), consistent with the overall structure of PF09361 Pfam-group phasins. Studies in other bacteria characterize the major biomechanical role of phasins as acting as a biosurfactant to coat hydrophobic carbon and energy storage granules composed of various polyhydroxyalkanoates (PHA)–with polyhydroxybutyrate (PHB) being the major

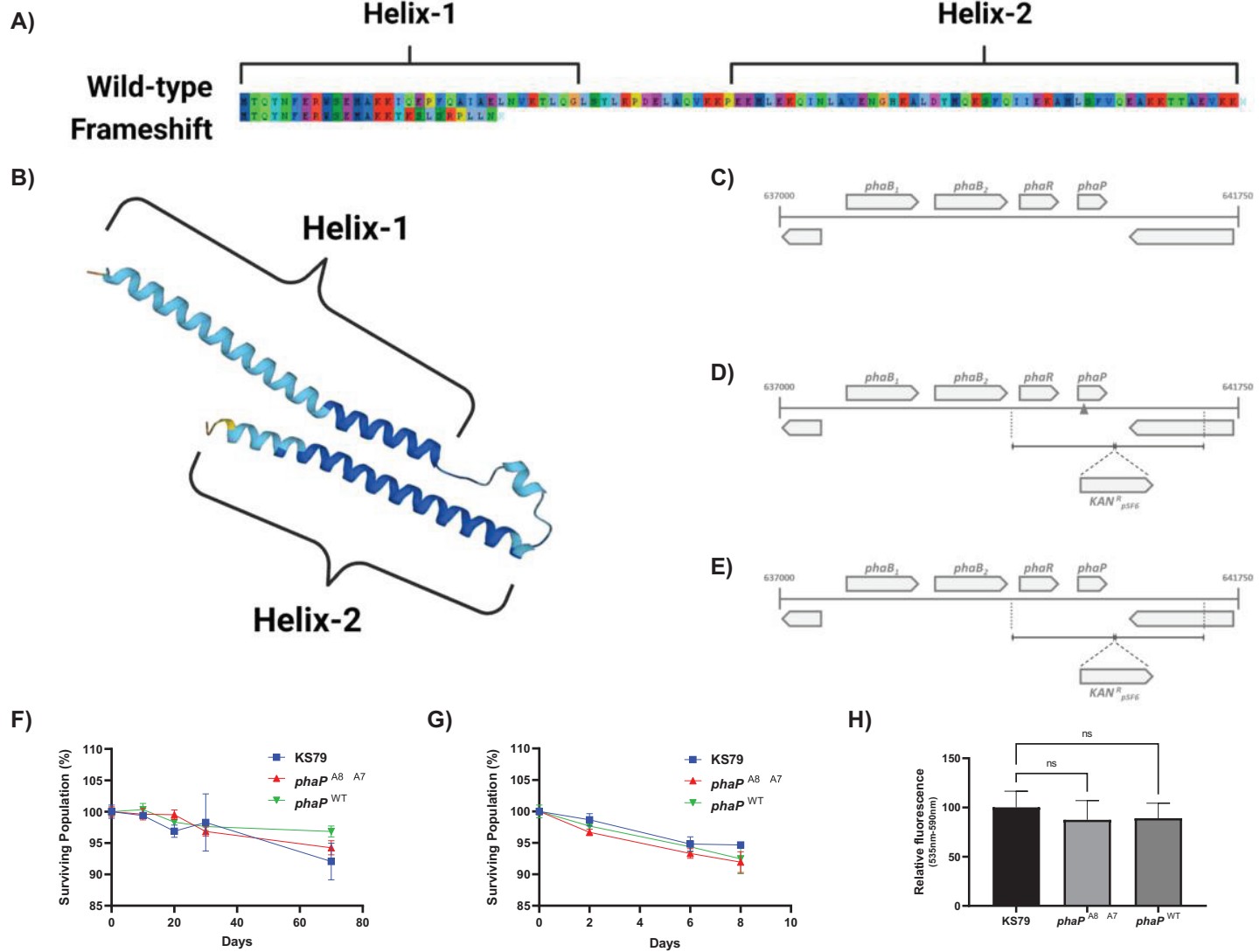

**Figure 3 Genetic and phenotypic characterization of wild-type and mutant alleles of *phaP* in *L. pneumophila*.** (A) Amino acid sequence of *L. pneumophila* str. Philadelphia-1 PhaP (above), annotated with predicted α-helices 1 and 2 and predicted amino acid sequence of the *L. pneu-mophila phaP*^A8→A7 gene product (below), showing a divergent sequence after 15 amino acids and pre-mature truncation of the translated product. Image produced using AliView 1.28 (*Larsson, 2014*). (B) AlphaFold-predicted (*Jumper et al., 2021*; *Varadi et al., 2022*) structure of PhaP (UniProt: Q5ZY12), showing assembly of two α-helices 1 and 2 and linker region. (C–E) Gene organization around the *phaP* locus including *phaB₁*, *phaB₂*, *phaR*, and *phaP* as an operon on the sense strand. D and E show the mutant construction strategy, with the mutated allele shown in the 5' flanking region of the Kan^R cassette. Survival of starvation in Fraquil of indicated strains at (F) 37 °C or (G) 42 °C shows no significant differences between wild-type KS79 and the constructed mutants, $n = 3$. (H) Polyhydroxybutyrate production evaluated with Nile red staining. Fluorescence was measured with excitation at 535 nm and emission at 590 nm normalized to mean fluorescence in KS79. Significance is depicted relative to KS79, $n = 3$, Data shown represents mean +/− SD.

PHA in *L. pneumophila* (*James et al., 1999*; *Gillmaier et al., 2016*). Proteinaceous accumulation of phasins on the surface layer of these lipids is a factor influencing the ultimate balance between the size and number of granules within each cell, as well as the final biomass yield (*Kuchta et al., 2007*).

The $A_8 \to A_7$ frameshift observed in HA-1 and HA-2 causes a premature truncation of the protein within the first N-terminal helix (Fig. 3A). Because phasins coat free PHB

granules through a hydrophobic interaction, it is plausible that the N' 15 residues left unchanged by this mutation are sufficient to maintain an effective separation between these lipid bodies and preserve the regulation of PHB metabolism. The maintenance of *L. pneumophila* integrity during long-term starvation in water systems involves its proper mobilization and consumption of intracellular PHB stores. Testing for whether this phasin mutation affects resistance to starvation independent of its effects on heat resistance, we compared the survival curves of *L. pneumophila* strains containing the wild-type and the mutated phasin allele with their KS79 ancestor strain (Figs. 3C–3E) for survival in Fraquil in suspension at both 37 °C and 42 °C (Figs. 3G–3H). There were no observed differences in the loss of culturability under these conditions, suggesting the conservation of PHB regulation and metabolism even in the phasin frameshift mutant. This macrophenomenon was reflected in spectrophotometric fluorescence measurements of populations stained with Nile red—a lipophilic dye used to quantify intracellular PHB content (*James et al., 1999*; *Alves et al., 2017*)—which showed no significant between-strain differences (Fig. 3F). If *L. pneumophila*'s phasin protein–like its orthologues in other bacteria–is involved in the liquid-liquid phase separation of PHB granules in the bacterial cell volume, then these data collectively suggest that truncation of PhaP to its N-terminal fragment does not quantifiably disturb PHB regulation in this system.

## Transcriptomic effect of PhaP frameshifting

Previous RT-qPCR analysis in our adaptive laboratory evolution model showed transcriptomic changes in the heat-adapted lineages which caused a sustained increase in heat shock gene expression levels even at ambient non-heat stress temperatures (*Liang, Cameron & Faucher, 2023*). With our isolated *phaP* frameshift mutant, as well as a negative-control allelic exchange mutant containing the wild-type allele of *phaP*, we compared the magnitude of the heat shock response as quantified by changes in the expression of *dnaJ*, *dnaK*, and *rpoH* (Figs. 4A–4C). RpoH is, as a sigma factor, a global regulator of the bacterial heat shock response, while DnaK and DnaJ are a chaperone pair that were also mutated in the HA-1 and HA-2 lineages. In each case, we found significantly elevated up-regulation relative to KS79 in the strain carrying $phaP^{A8 \to A7}$ but not in the strain carrying wild-type *phaP*. In our initial genetic screen, *phaP* was grouped in an evolutionary trajectory with *mreC* and *rodA* mutations seen in lineage HA-2 (Fig. 1A), two genes associated with cell wall structure and for whom mutations accumulated on a $phaP^{A8 \to A7}$ background made no additional contributions to heat shock resistance (Fig. 2B). When mutations from populations HA-1 and HA-2 in *dnaK* and *dnaJ* were applied to a $phaP^{A8 \to A7}$ background, however, a pattern of epistatic interaction emerged (Fig. 4D). The heat resistances of single mutant $phaP^{A8 \to A7}$ and double mutant $phaP^{A8 \to A7}$, $dnaJ^{F95Y}$ were statistically indistinguishable, and both were higher than that of the wild type. Double mutants of $phaP^{A8 \to A7}$ with either allele of *dnaK*–M94I with or without V373L–were much less robust in the face of heat shock than the *phaP* single mutant with heat resistance measures statistically undifferentiated from those of the ancestral strain. The significant deterioration of heat tolerance seen in both *phaP*, $dnaK^{V373L, M94I}$ double mutants was relieved by the addition of $dnaJ^{F95Y}$ (Fig. 4D, open symbol), implying

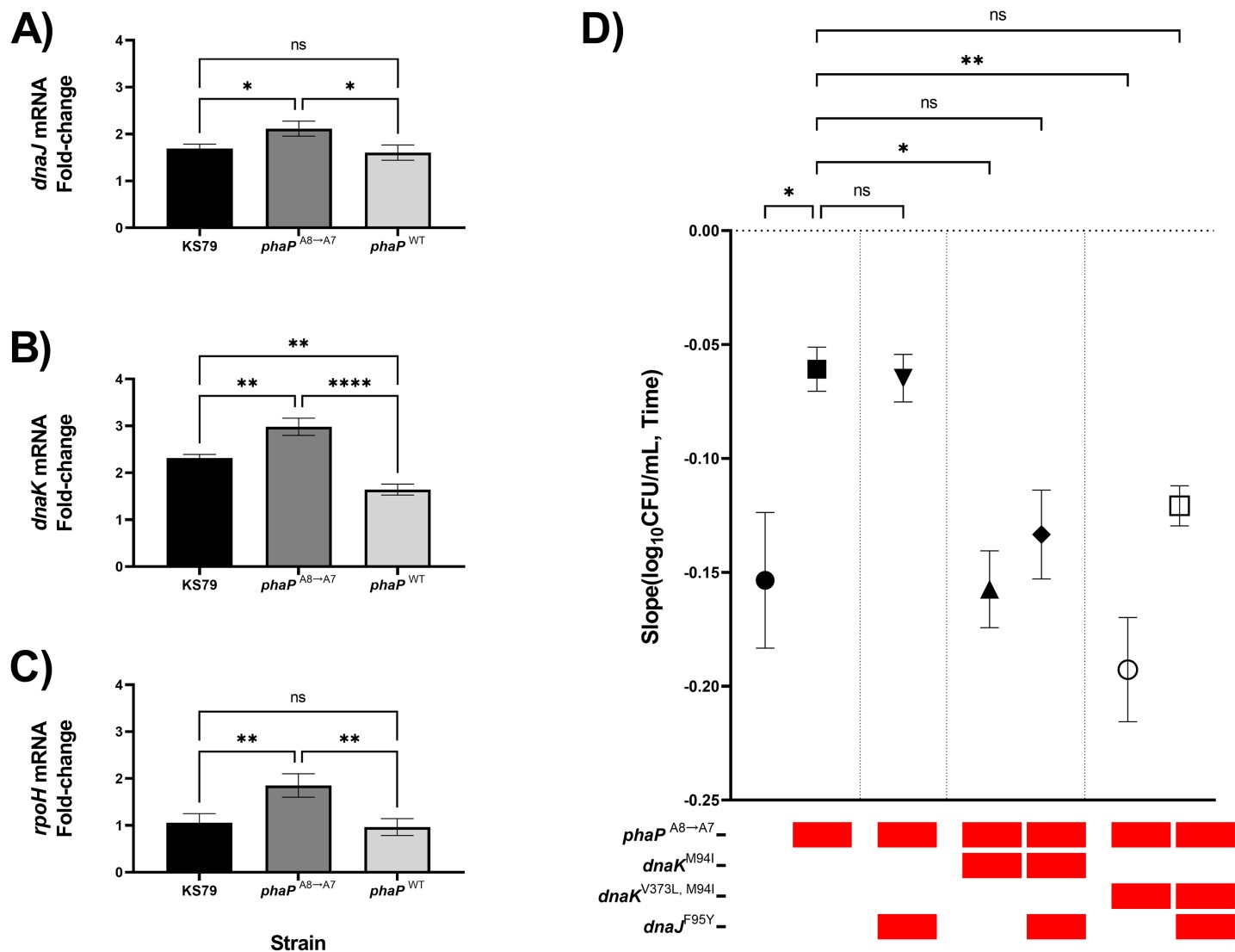

**Figure 4  phaP frameshift affects the transcriptional and phenotypic heat shock response.** (A–C) Transcriptional changes in (A) *dnaJ*, (B) *dnaK*, and (C) *rpoH* mRNA levels following a 5 min exposure to heat shock were measured by RT-qPCR and quantified by efficiency-corrected ddCt (*Yuan, Wang & Stewart, 2008*). Data shown represent mean +/− SEM and significance calculations show two-tailed one-way ANOVA with Tukey's correction for multiple comparisons, $n = 3$ ($^*p < 0.05$; $^{**}p < 0.01$; $^{***}p < 0.001$; $^{****}p < 0.0001$). (D) Comparisons of thermal kill rate at 55 °C between wild-type KS79, single mutant, double mutant, and triple mutant identified in the below key. Data shown represent mean +/− SEM and significance depicts ANCOVA with Šidák correction for multiple comparisons ($^*p < 0.05$; $^{**}p < 0.01$; $^{****}p < 0.0001$).

underlying interactions between the physiological effects of this phasin frameshift and the chaperone network central to the heat shock response of *L. pneumophila*.

## DISCUSSION

Consistent with other adaptive laboratory evolution studies using bacteria as an experimental platform, there was an overall linear accumulation of mutations in *L. pneumophila* heat-adapted lineages over the selective passages (*Good et al., 2017*; *Lenski, 2017*). During the evolution, no changes in mutation rate were observed in any of the three

lineages and no candidate mutations which could plausibly effect a hypermutator phenotype were observed (*Sniegowski, Gerrish & Lenski, 1997*). Inspection of sequencing data did not show slow shifts in allele frequency or the co-occurrence of stable sub-populations suggesting that our experimental lineages were not clonal. One significant divergence between *L. pneumophila* and some well-known systems of experimental evolution in bacteria is its demonstrated natural competence (*Stone & Abu Kwaik, 1999*; *Sexton & Vogel, 2004*). Several triggers are observed to underly the presentation of competence in the species including growth phase, UV damage, and antibiotic stress (*Sexton & Vogel, 2004*; *Charpentier et al., 2011*). Theoretical and experimental insights into the distribution of fitness effects of random mutations treat the likelihood of detrimental mutations as being much higher than the likelihood of beneficial mutations (*Robert et al., 2018*). Except for rare coincident back-mutations, these detrimental mutations accumulate as a genetic load in clonal systems. In the absence of recombination, negative mutations accumulate onto clonal genomes and act as a drag on gains in population fitness. Synthetically-recombinant *E. coli* have been used to experimentally confirm theoretical predictions that the integration of external DNA sources can relieve these deleterious effects and speed the rate of mutation fixation in a population (*Peabody et al., 2016*). In our evolutionary system, it seems that recombination promoted by DNA stress during exposure to heat shock or the transition from exponential to post-exponential phase growth would have acted to hasten the sweep of adaptive mutations acquired from the extracellular DNA contributions of dead cells through the rest of the population (*Peck, 1994*). Seeded from a single ancestor and propagated in axenic culture, there was no standing variation at the outset of our adaptive laboratory evolution nor was there any genetic input from other strains of *L. pneumophila* or from other microbial species, as would be expected in a *bona fide* hot water distribution system (*Ji et al., 2018*). Genomic analysis of the *Legionella* genus shows repeated introgression and co-option of eukaryotic genes, a telling sign of the significance of recombination on the evolution of the species (*Gomez-Valero et al., 2019*).

The biothermodynamic interactions precipitated at higher temperatures are complex and highly-networked, ultimately impacting all major compartments of the bacterial cell (*Baatout, De Boever & Mergeay, 2005*). Traditional understanding of the heat shock response focuses on the resolution of protein damage through chaperone-guided refolding or disaggregation and the proteolytic clearance of irreversibly-altered polypeptides (*Feder & Hofmann, 1999*; *Katikaridis, Bohl & Mogk, 2021*; *McGuire & Nano, 2023*). This is reflected in five of the genes analyzed in this manuscript (*htpG, dnaJ, dnaK, clpB,* and *clpX*) which are, variously, bacterial chaperones, co-chaperones, and disaggregases. Numerous mutations in these genes were seen in the evolutionary system, not all of which were preserved through to the 70-passage conclusion of the model. Collated across the three lineages, these include DnaJ$^{P334L}$, DnaJ$^{R27W}$, and DnaJ$^{W336R}$; DnaK$^{A208T}$; HtpG$^{A43V}$, HtpG$^{Q427*}$, and HtpG$^{S101F}$; ClpB$^{R188G}$; and ClpX$^{G204S}$. Motivated by a desire to maintain a strong selective pressure on our increasingly heat-adapted lineages, the temperature was raised three times over the course of our experiment from the initial 55 °C step-wise at 20-passage intervals to the final temperature of 59 °C. Thus, these lineages were exposed to

four graded macroenvironments during the time-transect of the adaptive laboratory evolution, each of which would have interacted with the specific genotypes present to generate distinct cellular environments of misregulated proteins, membranes, and other cellular subcompartments. Four mutations in heat shock genes (DnaJ$^{P334L}$, DnaK$^{A208T}$, DnaJ$^{R27W}$, and HtpG$^{S101F}$) were abruptly swept from the populations at the passage 30–35 interval that marks the discontinuous transition from 57 °C to 58 °C exposure. While it is possible that these variants could have been naturally outcompeted by other alleles given sufficient exposure to the milder heat shock environment, their concerted disappearance at this threshold suggests that there are peculiarities in the toxic effects of heat shock at even subtly different temperatures. Translationally, this emphasizes the importance that temperature has on the evolutionary trajectory of *L. pneumophila* and underscores the influence of maintaining well-adjusted hot water systems both during day-to-day use and when remedial pasteurization is required.

Direct genetic manipulation of *L. pneumophila* allowed us to isolate and measure the direct effect of variant alleles fixed in our evolved populations. Considering single mutations to separate out the effects of epistatic interactions, we observed that mutations in *dnaJ* and *htpG* were significantly beneficial with respect to survival time, as was a double mutation *dnaK*$^{M94I, V373L}$. The single mutation of greatest effect size was in a previously unannotated protein Q5ZY12 (*lpg0563*, A9F03_02970), predicted to code for a phasin. As observed before, these mutations were not observed in environmental or clinical isolates suggesting possible pleiotropies in the built environment with competing parameters in a complex natural system that favour alternative sequence variants (*Liang, Cameron & Faucher, 2023*). We do note, however, that isolates from hot water systems are relatively rare in published genome databases which are often dominated by strains of clinical or other environmental sources—such as cooling towers—and therefore largely reflect alternative evolutionary scenarios.

To test for intra-trajectory epistasis between co-mutated genes, we tested the linearity of heat tolerance acquisition in the four independent lineages. Ordinary least squares linear regression reported significantly positive relationships in TRA-1 and TRA-4 (Figs. 2A and 2D) between the number of mutations and TKR$_{55 °C}$, with goodness of fit $r^2 = 0.916$ and $r^2 = 0.618$ respectively. By contrast, mutations accumulating in trajectories TRA-2 and TRA-3 (Figs. 2B and 2C) did not show the same linear response. Mutations *mreC*$^{V266A}$ and *rodA*$^{R112S}$ in genes related to cell wall synthesis were neutral with respect to heat shock tolerance when added on the *phaP*$^{A8→A7}$ background in TRA-2 (Fig. 2B). Comparisons between TRA-1 and TRA-3 (Figs. 2A and 2C) which share a coincident mutation *dnaJ*$^{F95Y}$ differ in the behaviour of the third mutation; *clpX*$^{G202A}$ is beneficial in the context of TRA-1 and *clpX*$^{E203D}$ is detrimental in the context of TRA-3. While the *clpX*$^{E203D}$ mutation appears deleterious in this limited, synthetic context when aggregated with its indirect collaboators *dnaJ*/*dnaK* (*Doyle, Hoskins & Wickner, 2007*; *Laut et al., 2022*), its maintenance in HA-2 under strong selection suggests that other mutations in the evolutionary background of the lineage are compensatory, or that it provides off-target advantages in this experimental system.

Extending this observation, we computed per-codon $d_N/d_S$ with genomegaMap to predict the regions of each gene under greater or lesser selection pressure. None of the tested genes had an estimated overall mean ω greater than one, although $P(\omega \geq 1) > 0.025$ for a subset of codons distributed across the individual genes in the sliding-window model. The relatively labile codons were significantly enriched in non-heat shock genes, suggesting that purifying selection maintains sequence conservation in the chaperones and disaggregases. Such conservation has been observed in bacteria and other systems and is generally attributed to the vital importance of protein quality control mechanisms in buffering the predominantly negative structural variations introduced by the typical *de novo* nucleotide-level change (*Mack & Shorter, 2016*; *Agozzino & Dill, 2018*; *Alvarez-Ponce, Aguilar-Rodríguez & Fares, 2019*). These data also accorded with findings that core genes disproportionately accumulated non-synonymous mutations during adaptive laboratory evolution, while being more conserved in environmental samples (*Maddamsetti et al., 2017*).

There was no overall correlation between per-codon ω and the location of mutations observed in the three heat-adapted lineages, with the exception of *clpX*. *In vivo*, hexameric ClpX threads substrates into the heptameric ClpP peptidase to unwind and degrade ssrA-tagged cellular proteins and translationally-stalled polypeptides (*LaBreck et al., 2017*). This AAA+ motor protein experienced three mutations—G202A, E203D, and G204S—although only the first two eventually fixed in a population. These three adjacent residues sit in a stretch of higher estimated ω which corresponds to the residues linking the pore-2 loop and the RKH loop subdomains in the ClpX polypeptide, a region well-conserved between *L. pneumophila* and *E. coli* (*Fei et al., 2020*). The pore-2 and RKH loop subdomains of ClpX are involved in substrate recognition and translocation through the axial pore of the ClpX multimer into the peptidase chamber of ClpP (*LaBreck et al., 2017*; *Fei et al., 2020*). Cryo-EM structures of the ClpXP assembly model residues from these two loops in direct contact with the translocated substrate (*Fei et al., 2020*), suggesting that the variation seen in our adapted lineages and the $d_N/d_S$ analysis could be related to altered substrate targeting between different environments.

With numerous well-validated heat shock proteins acquiring mutations during the adaptive laboratory evolution, it was a surprise that the single tested mutation with greatest observed effect size was a frameshift in *phaP*. Sharing a phasin_2 domain and gross structure with other members of Pfam PF09361 (*Pötter et al., 2004*), this polypeptide appears to be the sole phasin encoded in the *L. pneumophila* str. Philadelphia-1 genome. Arranged on the chromosome as the final gene in a four-gene operon (Fig. 3A) encoding *phaB₁*, *phaB₂*–two gene copies of an acetyl-CoA reductase involved in PHB synthesis–, and *phaR*–a transcriptional repressor of phasin expression, *phaP* encodes a small 11.3 kDa protein comprised of two long α-helices separated by a flexible linker. Similar phasin_2 family proteins, including those expressed by *Azotobacter* sp. FA8 or *C. necator*, are larger with a greater number of α-helices (*Pötter et al., 2004*; *de Almeida et al., 2011*). Experimental and computational data suggest that these proteins are able to multimerize through coiled-coil interactions (*Pfeiffer & Jendrossek, 2013*; *Mezzina et al., 2014*), although these results are not known to extend to the *L. pneumophila* protein.

Tight regulation of metabolic activity is fundamental to *L. pneumophila*'s ability to endure a life cycle that periodically alternates nutrient uptake and replication inside eukaryotic hosts with starvation and transmission as a planktonic bacterium (*Sahr et al., 2017*). Consequently, the genome encodes sophisticated regulatory networks to control this phase switch, including numerous avenues of cross-talk to dial in expression *via* pre- and post-transcription mechanisms (*Dalebroux et al., 2010*; *Sahr et al., 2017*; *Mendis et al., 2018*). A number of these systems, associated with the ability of *L. pneumophila* to adapt to a stressful nutrient-deprived freshwater ecosystem, have also been shown to influence its survival to heat stress (*Li et al., 2015*; *Mendis et al., 2018*). Another strategy adopted by *L. pneumophila* to promote its long-term survival in water lies in the accumulation of PHB, a classic bacterial strategy to silo away carbon and free energy in times of plenty and consume them in times of lean (*James et al., 1999*; *Gillmaier et al., 2016*). Storage of this lipid polymer is linked with the accrual of PHB-associated proteins, notably a phasin shell which is hypothesized to coat the surface of individual small PHB granules and prevent their agglomeration into larger, enzymatically-inaccessible globules (*Mezzina & Pettinari, 2016*).

Our *L. pneumophila* mutants carrying the phasin frameshift mutation had an increased heat shock resistance without deficits in starvation survival or accumulation of PHB. Speculatively, it is possible that the portion of the protein preserved by this mutation is sufficient to carry out the greater part of a phasin's physiological role. Another possibility lies in the nature of the mutated homopolymeric poly($A_8$) tract. Comparative genomic methods have shown an enrichment of these tracts proximal to the 5' regions of prokaryotic genes, leading *Orsi, Bowen & Wiedmann (2010)* to propose the view that these represent a generalized regulatory mechanism on an evolutionary timescale. Because these sites are susceptible to slipped strand mispairing, they offer a reversible adaptive opportunity upstream of transcriptional regulation for genomically-encoded phase switching between functional products and truncated polypeptides. Notably, this same poly(A) region was independently mutated in another heat-adapted lineage in our experimental system showing its relatively high mutability. To our knowledge, this poly(A) tract is not conserved in *phaP* sequences outside of *Legionella*, suggesting that this could represent an adaptive mechanism specific to this genus.

Subcellular connections between PHB granules and other compartments of the bacterial cell contribute to the surprisingly diverse effects of the PHB-phasin carbon storage system. Monomeric and oligomeric degradation products of PHB–liberated from storage granules by the actions of depolymerase *phaZ*–have antioxidant behaviours as powerful scavengers of free radicals and appear to protect the bacterial cell from numerous stressors in addition to starvation (*Wang et al., 2009*; *Obruca et al., 2016*; *Koskimäki et al., 2016*; *Alves et al., 2020*). Heat shock, a non-specific insult to the bacterial cell, has been shown in the enterobacterium *E. coli* to increase cellular levels of reactive oxygen species with recovery being enhanced in the presence of glutathione (*Marcén et al., 2017*, *2019*). Destabilization of the PHB granule secondary to defects in the phasin shell could lead to physical instabilities in the lipid particle or to its attack by other granule-associated proteins during heat shock, liberating free monomers or multimers into the cellular milieu and dampening

radical chain reactions. Furthermore, the heterologous expression of *phaP* from *Azotobacter* sp. FA08 in recombinant *E. coli* was associated with increased stress resistance in both PHB-accumulator and non-accumulator strains (*de Almeida et al., 2011*). This was proposed by the authors to reflect *bona fide* direct chaperone activity of the phasin protein within the cell. In this system, PHB production was found to increase the expression of heat shock genes such as *groEL/ES*, *dnaK*, and *ibpA* while PhaP production was found to lower expression of the same genes. Follow-up *in vivo* and *in vitro* work showed that the phasins in this system physically associated with aggregates of misfolded proteins and that the purified phasin could slow the misfolding of an aggregation-prone model polypeptide (*Mezzina et al., 2015*). While the essential nature of the frameshifted *phaP* polypeptide in our system is unclear, collectively our data describe a model of direct or indirect interactions between *phaP* and the *dnaK/dnaJ* chaperone system. This inference is based on the observed divergence between the positive effects of the single mutations in isolation and the negative epistasis when combined onto one chromosomal background. A proposed mechanism can be imputed by analysing the surface proteome attached to PHB granules, which is seen to include heat shock proteins GroEL, DnaK, IbpA, and IbpB (*Lee, 2006*; *Mezzina & Pettinari, 2016*). Our RT-qPCR data is consistent with a model of chaperone titration where the phasin defect exposes the hydrophobic surface of the PHB granule, causing it to sequester chaperones from the cellular store of heat shock proteins (*Mezzina & Pettinari, 2016*). This mimics the protein misfold pathway of heat shock response induction, liberating the heat shock-responsive sigma factor RpoH from its chaperone-mediated destruction by the membrane protease FtsH and freeing it to alter the global transcription patterns of the RNA polymerase holoenzyme (*Gamer et al., 1996*; *McCarty et al., 1996*). Somewhat curious is the question of why the carriage of DnaJ$^{F95Y}$, DnaK$^{V94I}$, and DnaK$^{V373L, V94I}$ in conjunction with the *phaP* frameshift was favoured in lineages HA-1 and HA-2, when we observed negative epistasis in our synthetic constructs. Speculatively, we could imagine that the shifting environment of increasing temperatures has canalized development onto a track that favours the acquisition of these mutations over alternatives, or that these mutations have off-target effects that increase growth and survival or decrease lag time of the *L. pneumophila* populations in the interstitial ranges between heat shocks. It is also probable that there are interactions with mutations in the remaining untested background of accumulated mutations in the heat-adapted populations, owing to the observations that the lineages HA-1 and HA-2–while expressing negatively-epistatic mutations in *dnaK*, *dnaJ*, and *phaP*–are clearly capable of tolerating strong heat shocks (Fig. 2).

## CONCLUSIONS

In tandem with the construction of water infrastructure, the environmental pathogen *L. pneumophila* has found an expanded range of novel habitats. Outbreaks, particularly those causing nosocomial infections, are commonly associated with the contamination of hot water distribution systems. We report further results from an experimental system to recapitulate the long-term evolutionary response to heat shock of populations under serial exposure to incomplete pasteurization efforts. We find that heat shock resistance is a

complex trait with the majority of mutations selected for in evolved populations contributing small additive benefits. A surprisingly large effect size mutation was found which epistatically links the classic chaperone proteins of the heat shock response to long-term carbon/energy storage systems of the bacterium. Speculatively, the heat shock response in *L. pneumophila* may have been co-opted as a pleiotropic participant in the response to other proteotoxic stressors during its natural biphasic switch between hostile eukaryotic host and nutrient-sparse extracellular environment. Future work may focus on the evolutionary response of *L. pneumophila* to other common disinfectants of engineered water systems.

## ACKNOWLEDGEMENTS

We thank the Digital Resource Alliance of Canada for computational resources to support the sequencing and analysis pipelines in this manuscript.

### Funding

This work was supported by a Natural Science and Engineering Research Council of Canada Discovery Grant (RGPIN/04499–2018) to Sebastien P. Faucher. Jeffrey Liang was the recipient of a Canada Graduate Scholarship—Master's award from the Natural Science and Engineering Research Council of Canada. The funders had no role in study design, data collection and analysis, decision to publish, or preparation of the manuscript.

### Grant Disclosures

The following grant information was disclosed by the authors:
Natural Science and Engineering Research Council of Canada Discovery: RGPIN/04499–2018.
Natural Science and Engineering Research Council of Canada.

### Competing Interests

The authors declare that they have no competing interests.

### Author Contributions

- Jeffrey Liang conceived and designed the experiments, performed the experiments, analyzed the data, prepared figures and/or tables, authored or reviewed drafts of the article, and approved the final draft.
- Sebastien P. Faucher conceived and designed the experiments, authored or reviewed drafts of the article, and approved the final draft.

### Data Availability

The spectrophotometry, bacterial CFU count, RT-qPCR data, and a genome list for dN/dS computation raw data is available in the Supplemental Files.

The Illumina short-read data are available at National Centre for Biotechnology Information's Sequence Read Archive: PRJNA994942.

## Supplemental Information

Supplemental information for this article can be found online at http://dx.doi.org/10.7717/peerj.17197#supplemental-information.

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
