# Peer review of "Interactions between chaperone and energy storage networks during the evolution of Legionella pneumophila under heat shock"

_PeerJ, doi:10.7717/peerj.17197_

## Round 0.1 · original submission · Major Revisions

Dear Drs. Liang and Faucher:

Thanks for submitting your manuscript to PeerJ. I have now received two independent reviews of your work, and as you will see, the reviewers raised some concerns about the research. Despite this, these reviewers are optimistic enough about your work and the potential impact it could have on research studying Legionella biology. Thus, I encourage you to revise your manuscript, accordingly, considering all the concerns raised by both reviewers.

Please address the concerns about your experimental design, either by conducting more analyses as suggested by the reviewers or arguing why these additional analyses are not needed.

Please ensure that rationale is provided for dataset inclusion and parameters, and that your work is robust enough to support your conclusions.

The reviewers provide many suggestions to help improve your work and manuscript (especially grammar and presentation).

Therefore, I am recommending that you revise your manuscript, accordingly, considering all the issues raised by the reviewers.

Good luck with your revision,

-joe

Reviewer 1 ·

Basic reporting

Liang et al., present a good job in the evolution of Legionella pneumophila (LP) under heat
shock and found some special mutations in the chaperone-related genes that might help LP combat the thermal environment. The experimental design is reasonable, and there is no problem with interpreting the data. The article is well written and structured, but still, some concerns of mine could be raised.

Experimental design

The experimental design is reasonable, and there is no problem with interpreting the data.

Validity of the findings

Liang et al., present a good job in the evolution of Legionella pneumophila (LP) under heat
shock and found some special mutations in the chaperone-related genes that might help LP combat the thermal environment.

Additional comments

1. Through the article, we learned that some special mutations in the chaperone-related genes may contribute to the adaption of LP to the heat environment. This is gained from and verified by the experiment of adaptive laboratory evolution. However, are there any mutations mentioned in the article that could also be found in the artificial environmental strains (e.g., those LP isolated from the hot tap)? I suggest the author view the NCBI genome database and make a comparison with those LP isolates from hot tap.
2. As a potential functional change in the chaperone during the adaptive laboratory evolution, is there any other function change of these isolates with special mutation? I think the growth rate in the medium and the mammalian cells, as well as the virulence of those strains with special mutations, should be examined to illustrate the mutation's impacts on public health.

Reviewer 2 ·

Basic reporting

This manuscript describes an interesting study on looking a heat-adapted evolution and survival of Legionella mimicking conditions found in hot-water plumbing. The design of the study, and interpretation of the findings are for the most part sound. The introduction section could benefit from editing to improve clarify and flow. The results section would benefit from use of subheadings, and clearer rationale for the approaches. The discussion was written well with insightful interpretation of the data and relevance to other studies. Some typos are noted for correction by the authors.

Experimental design

No comment

Validity of the findings

No comment.

Additional comments

1. Lines 99-101: note that the findings of Ge et al., 2019 relate to the CsrA orthology CsrR (lpg1593), rather than CsrA. Refer to Abbott et al. 2015 mBio Jun 9;6(3):e00595.
2. Lines 103-105: this statement seems out of context with the background information on heat shock response. Recommend that this statement be deleted or rephrased.
3. Lines 108-113: while informative, this section more suited to a literature review thesis chapter and should be deleted.
4. Lines 113-148: this section is a bit lengthy, particularly since some details are re-stated in methods section; please trim down.
5. Lines 171-172: were HA-1, HA-2, and HA-3 randomly chosen from the six lineages for this study?
6. Lines 208: given that complementation would not be useful in this study to negate artifacts, were KS79 isogenic mutants strains genome sequenced to verify absence of secondary mutations?
7. The Results section would benefit from the use of sub-headings to help guide readers throughout the findings.
8. Lines 281-283: Figure 1A and Figure 1B should be reverted to follow the chronological order in the text. Further, the findings presented in Figure 1A are very briefly mentioned with little to no explanation of the findings to the reader – for instance, for someone not familiar with population genetics, wouldn’t fixed mutations constantly increase rather than fluctuate? The temperature value shifts should be detailed in the legend for quick reference.
9. Lines 350-352 and Figure 2: could the authors provide the rationale for choosing 55°C as the temperature for thermal shock? This is particularly important since mutations were achieved after shifting to 57°C and higher. Additionally, in reference to the key below each graph, it is not clear why HA-# is matched with HA-# (HA-1 to HA-1, HA-2 to HA-2, HA-3 to HA-3) for heat tolerance?
10. Lines 396-399: it is not clear on the rationale for the selection of 37°C and 42°C as temperatures for the survivability assays which it is assumed that it is different from a heat tolerance assay? There are no details in the methods section. Why not choose a higher temperature? Do the thermodynamics change for truncated PhaP versus normal PhaP at different temperatures?
11. Lines 404-406: this finding is speculative. There is no direct evidence that shows Legionella PhaP coats PHB; this only inferred by orthologous function through studies on other bacteria.
12. RT-qPCR: the exposure conditions is 5 minutes at 55°C; would 10 minutes (in reference to the heat tolerance assay) have a greater impact on expression levels?
13. lines 573-575: is the truncation of PhaP most likely the result of transcriptional or translational frameshifting, or due to slipped strand mis-pairing? It seems to be the case for the latter rather than the former since genomic sequencing was done. The authors should clarify this.

Minor Comments
1. Line 56: correct typo
2. Line 377: correct typo
3. Lines 399-409: this statement appears to be incomplete

---

## Round 0.2 · Minor Revisions

Dear Drs. Liang and Faucher:

Thanks for revising your manuscript. The reviewers are very satisfied with your revision (as am I). Great! However, there are a few minor issues to attend to. Please address these ASAP so we may move towards acceptance of your work.

Best,

-joe

Reviewer 1 ·

Basic reporting

The author answered my concerns very well, the entire paper has a complete structure and the results have been adequately explained with relatively sufficient discussion. I think the article could be published.

Experimental design

The experimental design is reasonable, and there is no problem with interpreting the data.

Validity of the findings

It is a good job in the evolution of Legionella pneumophila (LP) under heat shock and found some special mutations in the chaperone-related genes that might help LP combat the thermal environment.

Reviewer 2 ·

Basic reporting

No comment.

Experimental design

No comment.

Validity of the findings

No comment.

Additional comments

The revision of the manuscript and accompanying figures are much improved. All of the major/minor comments have been satisfactorily addressed. No further response to minor comment #3 is required - the statement was misread by the reviewer.

It is recommended that the authors incorporate a concise version of the provided explanations to major comments #9, #10 and #12 in the text to the benefit of the readers' understanding and help convey the importance of the findings.

---

## Round 0.3 · accepted · Accept

Dear Drs. Liang and Faucher:

Thanks for revising your manuscript based on the concerns raised by the reviewers. I now believe that your manuscript is suitable for publication. Congratulations! I look forward to seeing this work in print, and I anticipate it being an important resource for research studying Legionella biology. Thanks again for choosing PeerJ to publish such important work.

Best,

-joe